# Spatial and Temporal Changes in Ecological Resilience in the Shanxi–Shaanxi–Inner Mongolia Energy Zone with Multi-Scenario Simulation

**Xinmeng Cai**, **Yongyong Song \***, **Dongqian Xue**, **Beibei Ma**, **Xianfeng Liu** and **Liwei Zhang**

School of Geography and Tourism, Shaanxi Normal University, Xi'an 710119, China; cxmdimon@snnu.edu.cn (X.C.); xuedq@snnu.edu.cn (D.X.); mabb@snnu.edu.cn (B.M.); liuxianfeng7987@163.com (X.L.); zlw@snnu.edu.cn (L.Z.)
* Correspondence: syy2016@snnu.edu.cn

**Abstract:** The energy-driven expansion of artificial surfaces has resulted in severe ecological problems. Scientific evaluation of regional ecological resilience under different scenarios is crucial for promoting ecological restoration. This study chose the Shanxi–Shaanxi–Inner Mongolia Energy Zone (SEZ) and modeled an ecological resilience evaluation based on resistance, adaptability, and recovery. Land-use change and ecological resilience from 1980 to 2020 were then analyzed. Moreover, the SEZ land-use patterns and ecological resilience in 2030 were simulated under business as usual (BAU), energy and mineral development (EMD), and ecological conservation and restoration (ECR) scenarios. The results showed that (1) the SEZ was dominated by cultivated land, grassland, and unused land. (2) Ecological resilience showed a changing trend of decreasing and then increasing, with high ecological resilience areas mainly located in the Yellow River Basin, whereas low ecological resilience areas spread outward from the central urban areas. (3) The ecological resilience level was the lowest under the EMD scenario and the highest under the ECR scenario. This study not only expands the analysis framework of ecological resilience research but also provides scientific support for ecological conservation in ecologically fragile areas with intensive human activity worldwide.

**Keywords:** ecological resilience; PLUS model; multi-scenario simulation; land use

## 1. Introduction

Energy development zones, as a crucial energy supply to ensure sustainable global socio-economic development, contain an abundance of energy resources, including oil, natural gas, coal, and renewable resources. Energy development dominates production and construction activities in these zones, and energy-based industries play a major role in regional economic development [1]. Energy development zones are one of the areas with the most intense human activities [2]. For a long time, the irrational development of energy and mineral resources has been accompanied by the piecemeal expansion of construction land, occupying a large amount of ecological space [3], which poses a serious threat to the ecosystems. Therefore, scientific evaluation and prediction of the ability of ecosystems in energy development zones to withstand risks is an effective measure to address the growing unsustainability problems.

The concept of ecological resilience characterizes the ability of ecosystems to absorb, resist, adapt, and recover from disturbance [4], and is of great significance in guiding energy development zones to promote ecologically sustainable development. Resilience was first introduced into the field of ecology by the ecologist Holling [4], and research achievements such as resilience concept identification [5,6] and indicators for resilience evaluation [7,8] have provided theoretical support for ecological resilience research. However, in practical application, it is even more important to construct a reasonable quantitative methodological framework for regional ecological resilience. Current research is focused on three

main aspects. Firstly, there is a focus on the impact of human activities on the ecological resilience process [9]. Research has shown that the expansion of construction areas and a development mode that purely pursues economic benefits can reduce the quality of ecosystem services [10]. Secondly, there have been studies combining the research methods of systems theory and ecology to evaluate the level of ecological resilience at different scales [11], explore its evolution process, and formulate development plans [12,13]. Lastly, attribution analysis of ecosystem change has been conducted to analyze the influencing factors of ecosystem change [14,15], and it was found that the level of urbanization, urban spatial patterns, and topographic conditions are key factors affecting ecological resilience. These studies provide a solid foundation and different perspectives for assessing ecological resilience.

As a hot research topic, the complex connotations of resilience have not been consistently finalized. For example, Macgillivray and Grime [16] believed that there may be a trade-off between resilience and resistance; Hodgson et al. [17] proposed that concepts of resistance and recovery can complement each other. Previous research [10,11] has illustrated that multiple processes influence resilience evaluation and that a single indicator cannot simply be used to measure resilience. This implies that it is necessary to establish appropriate evaluation frameworks based on different research objects. In 2019, Grafton et al. [18] proposed that social–ecological systems can be measured using the "3R model", which refines the dynamic process of resilience and has been initially explored in empirical case studies [19]. They explored resistance, recovery, and robustness as three important attributes of socio-ecological system resilience, and such definitions provide scientific and practical guidance on how different systems can achieve resilience. However, for energy development zones, which are vulnerable to human activities, the situation is somewhat different. It is crucial to focus on the adaptability of the ecosystems to disturbances [20] during ongoing activities like mining operations. Thus, this paper considers adding an "adaptability" index to modify the existing "3R" conceptual model to explore whether ecosystems can positively adapt to external disturbance by adjusting their internal processes when the disturbance continues to occur. In summary, this paper argues that the ecological resilience of energy development zones can be viewed as the ability of regional ecosystems to respond at different stages of disturbance. Ecological resilience is discussed comprehensively from three aspects: the resistance of the ecosystem in the transient or short-term period when the disturbance occurs, the adaptability during the continuation of the disturbance, and the recovery after the disturbance ends. These three are interconnected and indispensable and work together for ecological resilience, although their focus varies at different periods when disturbance occurs.

Some existing methods of constructing an ecological resilience evaluation index system through statistical data gradually show the drawbacks of different statistical calibers, missing data, and subjectivity [21,22]. In addition, most of the existing methods for measuring regional ecological resilience adopt the "scale-density-form" ecological resilience model based on administrative boundaries [21,23], which focuses on the impacts of human activities on the ecological environment, and less on natural factors such as land use, climate change, and topographic conditions. Land use, as a visual manifestation of the interaction between natural and human factors on the earth's surface [24,25], is an intrinsic driver of the evolution of ecological resilience in the SEZ [24], and land-use data are easier to access and collect than statistical data. Therefore, this paper uses land-use change as an endogenous drive to portray the changing law of ecological resilience. Furthermore, during the economic transformation and ecological civilization construction in energy development zones, conducting multi-scenario simulations can help clarify the future regional land-use expansion dynamics [26,27]. This, in turn, can offer valuable guidance in preparing for potential disturbances in regional ecosystems caused by uncertain risks [28]. Compared to traditional land-use simulation models [27,29,30], the PLUS model integrates the Land Expansion Analysis Strategy (LEAS) and CA model using a multi-class stochastic patch

seed. This integration results in a more precise and efficient land-use simulation model [31]. The paper chose the PLUS model as a tool to simulate trends in land-use changes.

The issue of declining ecological resilience based on land-use changes must be brought to the forefront of global attention, particularly in typical energy zones [32,33]. Energy development zones are confronted with the critical problem of uncontrolled expansion of construction land and degradation of ecological land. This creates a conflict between economic development and ecological protection [34,35], which must be regulated and optimized in advance. This paper focuses on the SEZ, a critical energy security base in China with a delicate ecological environment, as a case study site. The objective is to deepen the basic meaning of ecological resilience and establish a practical and adaptable assessment framework for ecological resilience based on land-use change in energy development zones worldwide. The results could offer initial insights into protecting ecological resilience and optimizing land use in ecologically fragile energy development areas worldwide. The potential links among resistance, adaptability, and recovery are explained and used to develop the ecological resilience evaluation model. Local spatial autocorrelation and the PLUS model are then used with the aim of (1) elucidating spatial and temporal changes in land use and ecological resilience in the SEZ from 1980–2020; (2) exploring spatial differentiation patterns of ecological resilience and the spatial agglomeration pattern of energy development enterprises in the SEZ in 2020; (3) simulating evolution patterns of regional land use and ecological resilience in 2030 under the three scenarios of business as usual (BAU), energy and mineral development (EMD), and ecological conservation and restoration (ECR).

## 2. Research Framework

### 2.1. Conceptual Framework for Ecological Resilience Evaluation in the SEZ

Ecological resilience is a dynamic process that is constantly changing and challenging to characterize with just one variable [17]. A conceptual breakdown of ecological resilience is crucial. This paper considers a comprehensive portrayal of the ecological resilience of the SEZ in terms of resistance, adaptability, and recovery and needs to elucidate the interconnections among these three aspects.

As shown in Figure 1, the focus on the ability of ecosystems to respond to external disturbances may differ at different stages. When ecosystems located in energy development areas respond to external disturbances such as mining operations and land expansion for construction, their ability to resist the disturbances in the transient or short term by relying on their conditions [17], such as the natural environment and biodiversity, is called resistance. In reality, ecosystems often require a longer period to respond to external disturbances to assess their ability to adapt to such disturbances. Contrary to episodic natural hazards like floods and earthquakes that threaten the region's ecological resilience, human activities such as mining operations, while constituting external disturbances to the SEZ's ecological resilience, are crucial for sustaining and driving local and broader economic benefits. Therefore, more attention needs to be paid to the ability of ecosystems to continuously adapt themselves as disturbances continue. This paper introduces the index of "adaptability" as an important aspect of ecological resilience evaluation based on the 3R model [18]. Ecosystems are considered resilient if they can adapt positively to external disturbances over an extended period [20]. The ability of an ecosystem to transition from a disturbed state to a stable state is referred to as recovery [19], which is a specific manifestation of ecological resilience after the end of external disturbances. Ecosystems in energy development zones may have the three abilities of resistance, adaptability, and recovery described above, but with different emphases, in the overall process of facing external disturbances. Energy development zones' ecosystems may have a higher chance of recovering to a stable state through human interventions like land reclamation and reforestation. However, this does not necessarily mean that their ability to resist risks initially and adapt to disturbances in response to risks is also high. Thus, a complete assessment of regional ecological resilience requires considering all three aspects. Unlike the early

prevention of natural disasters, this paper did not introduce the concept of early prevention into the SEZ ecological resilience evaluation model but expressed it by simulating future trends through multiple scenarios.

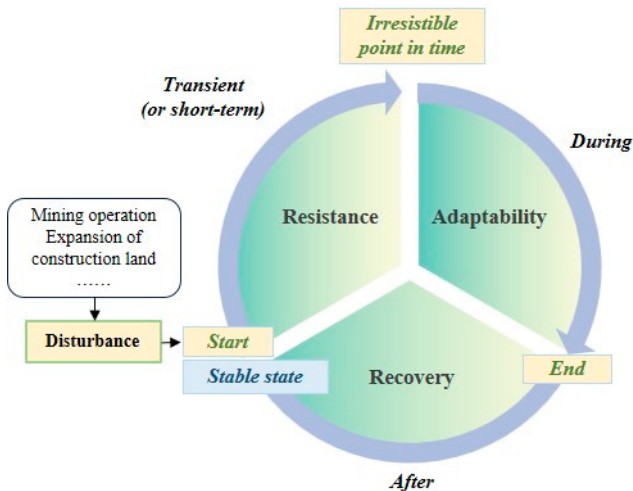

**Figure 1.** The conceptual framework for ecological resilience evaluation in the SEZ.

### 2.2. Framework for the Selection of Indices

Ecosystem resistance refers to the ability of an ecosystem to maintain its structural functions against short-term and transient external disturbances [17,36]. This aligns with the concept of ecosystem services. The benefits that ecosystems and ecological processes provide to sustain human survival and development are called ecosystem services [37]. From the perspective of human–nature coupling, healthy and resistant ecosystems should sustainably provide a range of valuable ecosystem services to fulfill human requirements while maintaining structural and functional integrity [38]. In the face of external perturbations like climate change, human activities, and natural disasters, these pressures can diminish both the quantitative and qualitative aspects of ecological services [39,40]. Consequently, ecosystems must be able to deliver sufficient and sustainable ecosystem services [41]. Ecosystems with limited services may become so fragile that they are unable to resist external disturbances and when disturbed struggle to provide ecosystem services and functions at the same level of value as before [40]. According to the Millennium Ecosystem Assessment, over 60% of ecosystem services are currently degraded. This degradation can lead to ecosystems losing the ability to support and protect themselves from external disturbances, which can ultimately pose a threat to both regional and global ecological security [42]. Hence, assessing the value of an ecosystem's services rendered at specific intervals over time can, to some extent, reflect the integrity of its structure and functioning, along with its resistance and ability to withstand risks.

This requires appropriate valuation of ecosystem services, which can be done using either value quantity or physical quantity assessments. Fixed-point observations facilitate the application of the physical quantity method in small-scale areas, yet obtaining data for large-scale areas poses a greater challenge [43]. Ecosystem service value (ESV) is a measure of the amount of value of ecosystem service functions using a monetization method [41], which is widely used in empirical studies due to its simplicity and practicality. The ecosystem service value approach was initially utilized in a study by Costanza et al. [44] for global ecosystems and biospheres, where the global ecosystem services' value was calculated. Then Xie et al. [45] developed a Chinese ESV equivalent factor table on this basis, which has been widely applied. Moreover, the ESV method, known for its ease of aggregating various services, proves suitable for comparative analyses of ecosystem service dynamics across different time frames and land-use scenarios. This paper employs ESV calculations to delineate ecosystem resistance.

Ecological adaptability is the ability of an ecosystem to continuously adjust itself in response to persistent external perturbations and internal demands [16,46]. The more stable the landscape structure of an ecosystem is, the more adaptable it is to external disturbances. The ecosystem landscape index was employed to characterize ecosystem adaptability, with landscape heterogeneity and connectivity reflecting the stability of the ecosystem landscape structure. Concerning these factors, the Shannon diversity index is sensitive to the non-equilibrium distribution status of each patch type in the landscape, and the area-weighted fractal dimension reflects the complexity of landscape patch shapes, both of which reflect landscape heterogeneity well; landscape fragmentation characterizes the degree of fragmentation in which the landscape has been segmented, and was used to measure landscape connectivity in this paper.

Ecosystem recovery reflects the ability and potential of an ecosystem to recover from a disturbed state to a stable state after an external disturbance has ended. Referring to the ecological resilience model and coefficients proposed by Peng et al. [38], this study believes that when an ecosystem encounters disturbances, unused land that has not been affected by human activities has a greater capacity for resilience, whereas the human-dominated construction land has a lower recovery and suffers from greater damage in the face of disturbance.

## 3. Materials and Methods

### 3.1. Study Area

The SEZ ($37°21'$–$40°16'$ N, $108°56'$–$111°29'$ E) is located in the middle and upper reaches of the Yellow River in China, at the junction zone of Shanxi and Shaanxi Provinces and the Inner Mongolia Autonomous Region, and covers a total of three provinces (autonomous regions), five cities, and 13 counties (districts, banners) (Figure 2).

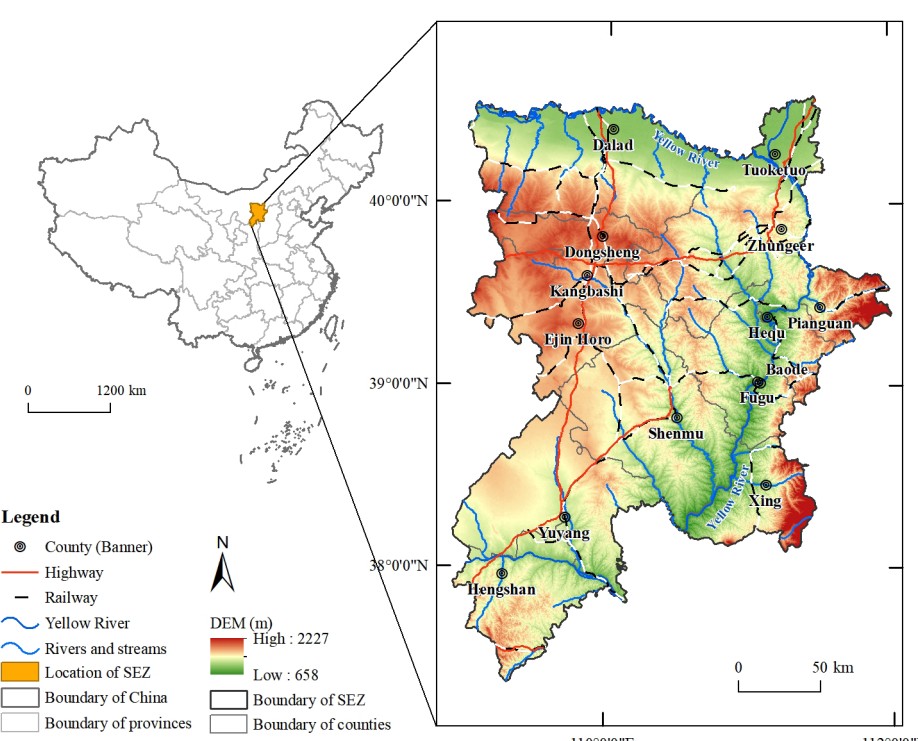

**Figure 2.** Location of the SEZ.

The SEZ is situated in the northern transitional zone of China, where agriculture and animal husbandry are intertwined, and consists of five major geomorphological types: loess hilly and gully areas, windblown sand areas, earth–rock mountainous areas, alluvial plains, and arid grassland areas. In addition, the SEZ is a key national soil erosion control

area and a key supervision area for soil and water conservation, with abundant reserves of energy and mineral resources, such as coal, oil, and natural gas. In 2020, the region had a population of 3.97 million, a GDP of CNY 6157.46 billion, and a construction land area of 2247.22 km². With intensive human activity and a prominent ecological status, the region is a national energy security guarantee base and an area sensitive to global climate change, making it an ideal case study for regional ecological resilience.

### 3.2. Data Sources

The research data in this paper are mainly land-use data, the driving factor data needed for the PLUS model, and other socio-economic data. The specific research data are shown in Table 1.

**Table 1.** Detail of all data.

| Category | Data | Years | Spatial Resolution | Data Sources |
|---|---|---|---|---|
| Land-use data | Land use | 1980–2020 | 30 m | CAS (https://www.resdc.cn/, accessed on 19 November 2023) |
| Environmental data | Annual average temperature | 2020 | 1000 m | CAS |
| | Annual average precipitation | | | |
| | Soil type | - | 30 m | CAS |
| | DEM | 2020 | 30 m | Geospatial Data Source (https://www.gscloud.cn/, accessed on 19 November 2023) |
| | Slope | 2020 | 30 m | Calculated from DEM data in ArcGIS |
| Socio-economic data | GDP | 2019 | 1000 m | CAS |
| | POP | 2019 | 1000 m | CAS |
| | Distance to primary road | 2020 | 30 m | OpenStreetMap (https://www.openstreetmap.org/, accessed on 20 November 2023) |
| | Distance to the secondary road | | | |
| | Distance to the tertiary road | | | |
| | Distance to railway | | | |
| | Distance to highway | | | |
| | Distance to open water | | | |
| | Distance to county (district/banner) governments | | | Baidu Maps (https://map.baidu.com/, accessed on 20 November 2023) |

1. Land-use data. The land-use data were obtained from the Resource and Environment Science Data Center of the Chinese Academy of Sciences (CAS). According to the classification rules formulated by the CAS [47], land use was classified into six types: cultivated land, woodland, grassland, water body, construction land, and unused land;

2. Driving factor data. When using the PLUS model for multi-scenario simulations, driving factors must be produced, including environmental factors and socio-economic factors. In addition to those shown in Table 1, the distance to the county (district, banner) administrative center and individual road levels were calculated using the ArcGIS 10.3 Euclidean distance tool;

3. Other socio-economic data. Grain sowing area and production data in the SEZ were obtained from the 2001–2021 Statistical Yearbooks of each county and district. Grain prices were obtained from the "National Compendium of Agricultural Product Costs and Benefits". Energy development enterprises in the SEZ were selected from the

2020 list of key pollutant discharge units by the ecological environment bureaus of each county (district, banner), and the coordinate information was obtained from the Baidu coordinate picking platform and transformed into WGS84 coordinates.

### 3.3. Methodology for the Calculation of Indices

Based on clarifying the connotation of ecological resilience in the SEZ, this paper referred to the relevant studies [48,49], and chose suitable calculation methods for each index to precisely depict the resistance, adaptability, and recovery of ecosystems in the SEZ.

### 3.3.1. The Resistance Index

The ESV equivalent factor table was corrected by using one standard equivalent value equal to 1/7 of the average grain yield market value in the SEZ [50]. Taking the average grain yield of 3351.03 kg·hm$^{-2}$ from 13 counties (districts and banners) in the SEZ from 2000 to 2020 as a substitute for the grain yield, and taking the national average grain selling price of CNY 2.26·kg$^{-1}$ from 2011 to 2020 as the grain price, the ESV equivalent factor in the SEZ was calculated to be CNY 1081.90·hm$^{-2}$, and the ESV of land-use types in the SEZ was obtained (Table 2).

**Table 2.** Coefficients of ESV of different land-use types in the SEZ (CNY·hm$^{-2}$·a$^{-1}$).

| Ecosystem Services | Ecosystem Sub-Services | Cultivated Land | Woodland | Grassland | Water Body | Construction Land | Unused Land |
|---|---|---|---|---|---|---|---|
| Supply services | Food production | 919.62 | 313.75 | 108.19 | 865.52 | 0 | 0 |
| | Raw material production | 432.76 | 714.05 | 151.47 | 248.84 | 0 | 0 |
| | Water supply | 21.64 | 367.85 | 86.55 | 8968.95 | 0 | 0 |
| Regulation services | Gas regulation | 724.87 | 2347.72 | 551.77 | 833.06 | 0 | 21.64 |
| | Climate regulation | 389.48 | 7032.35 | 1449.75 | 2477.55 | 0 | 0 |
| | Environmental purification | 108.19 | 2088.07 | 476.04 | 6004.55 | 0 | 108.19 |
| | Hydrological regulation | 292.11 | 5128.21 | 1060.26 | 110,613.46 | 0 | 32.46 |
| Support services | Soil conservation | 1114.36 | 2867.04 | 670.78 | 1006.17 | 0 | 21.64 |
| | Nutrient cycle maintenance | 129.82 | 216.38 | 54.09 | 75.73 | 0 | 0 |
| | Biodiversity protection | 140.65 | 2607.38 | 605.86 | 2758.85 | 0 | 21.64 |
| Cultural services | Aesthetic landscape | 64.91 | 1146.81 | 270.48 | 2044.79 | 0 | 10.82 |

The calculation formula for the SEZ ecosystem resistance is as follows:

$$Resi = \sum_{i=1}^{6} A_i \times VC_{ij} \tag{1}$$

where *Resi* is the ecosystem resistance index characterized by the ESV function. $A_i$ is the area of land-use type *i*, and $VC_{ij}$ is the *j*th ESV of land-use type *i*.

### 3.3.2. The Adaptability Index

Landscape heterogeneity and connectivity are equally important in describing ecosystem landscape structure, so their weights can be set as equal [38]. The formula is as follows:

$$Adap = 0.25SHDI + 0.25AWM + 0.5C \tag{2}$$

where *Adap* is the ecosystem adaptability index, *SHDI* is the Shannon diversity index, *AWM* is the area-weighted average patch fractal dimension, and *C* is landscape fragmentation, all of which were calculated using Fragstats 4.2 software.

### 3.3.3. The Recovery Index

The calculation formula for the SEZ ecosystem recovery is as follows:

$$Reco = \sum_{i=1}^{6} A_i \times RC_i \tag{3}$$

where *Reco* is the ecosystem recovery index, $A_i$ is the area of land-use type $i$, $RC_i$ is the resilience coefficient of land-use type $i$, which was determined with reference to Peng et al. [38].

### 3.3.4. The Resilience Index

The ecosystem resistance, adaptability, and recovery were normalized to [0,1] using the extreme deviation standardization method, and then the ecological resilience index was calculated. The formula is as follows:

$$Resilience = (Resi \times Reco \times Adap)^{1/3} \tag{4}$$

where *Resilience* indicates the ecological resilience index.

### *3.4. Spatial Autocorrelation Model*

The local autocorrelation LISA [51] was used to analyze the spatial distribution of ecological resilience and the energy development enterprise density agglomeration. The formula is as follows:

$$I = \frac{n(X_i - \overline{X}) \sum_{j=1}^{n} W_{ij}(X_j - \overline{X})}{\sum_{j=1}^{n}(X_j - \overline{X})^2} \tag{5}$$

where $n$ is the number of grids into which the SEZ is divided or the number of energy development enterprises; $X_i$, $X_j$ is the value of the ecological resilience or energy development enterprise kernel density for spatial locations $i$, $j$; $W_{ij}$ is the weight matrix of the adjacency relationship between geographical units; and $\overline{X}$ is the average of the ecological resilience value or the kernel density of energy-developing enterprises.

### *3.5. Land-Use Change Simulation Based on the PLUS Model*
### 3.5.1. PLUS Model

The PLUS model mainly consists of two parts: the LEAS and the CA model based on multiple random patch seeds (CARS) [31]; thus, the PLUS model can explore the driving factors of land expansion and better simulate the evolution of land-use patches. The LEAS module uses a random forest algorithm to sample the expansion portion of land use in different years and calculates the development probability of each land-use type and the driving factor contribution to the expansion of land use. The CARS module integrates random seeding and threshold-diminishing mechanisms to forecast future land-use distribution while considering development probability constraints.

In this paper, 14 driving factors (Table 1) were chosen for simulating land use in the SEZ in 2030. Moreover, water systems and nature reserves were set as restricted development areas. Before conducting the simulation, land-use distribution data for 2020 were simulated based on historical trends. In comparison with the actual land-use data in 2020, the results demonstrated high accuracy, with an overall accuracy of 89.2%.

### 3.5.2. Multi-Scenario Settings

Referring to the "Regulations on the Development and Construction of Soil and Water Conservation in the Border Region of Shanxi–Shaanxi–Inner Mongolia" and existing studies [10,26,52], in response to the actual situation of abundant mineral resources and

the severe soil erosion phenomenon in the SEZ, this study set up three types of land-use change simulation scenarios: BAU, EMD, and ECR.

1.  BAU scenario: Based on the actual development of the SEZ, according to the land-use change trend from 2010 to 2020, the area of each land-use type in 2030 was calculated using a Markov chain, which is the original 2030 prediction result generated by the PLUS model;

2.  EMD scenario: Since the implementation of the Western Development Strategy in 2000, the large-scale development of energy and mineral resources and the construction of supporting facilities in the SEZ have led to an accumulation of waste soil and slag, which has blocked rivers. In addition, coal mining has damaged the natural ecosystem structure and changed landscape patterns and geomorphology. Accordingly, this study identified a 50% increase in the probability of conversion of cultivated land, woodland, grassland, and water bodies to unused land. Additionally, a 30% increase in the probability of conversion of cultivated land, woodland, grassland, and water bodies to construction land was determined. Moreover, there was a 30% decrease in the probability of conversion of construction and unused land to cultivated land, woodland, grassland, and water bodies, and a 20% increase in the probability of conversion of construction land to unused land.

3.  ECR scenario: Under the promotion of a series of ecological restoration projects, such as returning farmland to woodland (grassland), the area of regional soil erosion has been significantly reduced, and the ecological construction results were remarkable. Therefore, this study designated the water system and nature reserve within the SEZ as a restricted development area. Simultaneously, it strictly limited the transfer of woodland, grassland, and water bodies, reducing the probability of conversion to construction and unused land by 50%. It also aimed to decrease the probability of cultivated land being converted to construction and unused land by 30% and increase the probability of unused and construction land being converted to woodland, grassland, and water bodies by 30%.

## 4. Results

### 4.1. Land-Use Change Characteristics

From 1980 to 2020, the land use in the SEZ was dominated by grassland (>48%), cultivated land (>23%), and unused land (>12%). During the 40-year study period, there was a significant increase in the area of construction land in the SEZ, with a rise of 156,332.34 hm². Notably, the construction land area remained relatively stable during the initial two decades but experienced a rapid expansion in the subsequent two decades. The figures highlight the swift expansion of construction land in tandem with China's rapid economic growth in the 21st century. Moreover, the area of ecological land has undergone significant changes in the past two decades. With the gradual implementation of economic development initiatives such as the "Western Development" policy, extensive deforestation and land clearing in the SEZ led to an 8.62% decline in woodland area from 2000 to 2010. Subsequently, from 2010 to 2020, as the Chinese economy transitioned from traditional rugged development practices to a greater focus on high-quality economic growth and ecological conservation, the woodland area exhibited a notable increase (Table 3 and Figure 3).

In this paper, a land-use transfer chord diagram was visualized by Origin for a more specific conversion relationship between each land-use type. There was no significant change in land-use transfer between 1980 and 1990. However, between 1990 and 2000, there was an expansion in the area of grassland. The largest transfer was observed from unused land to grassland, which accounted for 159,106.14 hm². As can be seen from the thickness of the chords, the interconversion between the three land types dominant in the SEZ was more pronounced and correlated during this decade. The period from 2000 to 2010 witnessed significant transformations, notably the noticeable transfer between construction land and other land-use types. The largest conversion occurred from grassland to construction land, amounting to 31,729.23 hm². Additionally, noteworthy conversions include cultivated

land to woodland (25,783.2 hm²) and to grassland (80,029.26 hm²), reflecting the emphasis placed by local governments on initiatives such as the "Returning Cultivated Land to Woodland (Grassland)" policy. In the period from 2010 to 2020, there was a sharp increase in the area of construction land, primarily converted from grassland (77,831.55 hm²) and cultivated land (21,482.28 hm²). Notably, the woodland area also expanded during this decade, predominantly converted from unused land.

**Table 3.** Land-use area and proportion in the SEZ from 1980 to 2020.

| Year | Index | Cultivated Land | Woodland | Grassland | Water Body | Construction Land | Unused Land |
|---|---|---|---|---|---|---|---|
| 1980 |  | 1,385,167.45 | 354,129.67 | 2,607,425.24 | 143,906.9 | 68,390.21 | 850,749.27 |
| 1990 |  | 1,384,814.49 | 355,127.44 | 2,594,057.03 | 138,313.8 | 69,435.17 | 868,012.62 |
| 2000 | Area (hm²) | 1,369,351.74 | 357,458.13 | 2,715,066.99 | 138,911.80 | 71,399.83 | 757,659.37 |
| 2010 |  | 1,332,817.65 | 326,925.14 | 2,828,018.35 | 127,356.69 | 130,247.36 | 664,509.28 |
| 2020 |  | 1,285,290.65 | 357,009.95 | 2,702,636.34 | 136,160.51 | 224,722.55 | 703,839.01 |
| 1980–1990 |  | −352.96 | 997.77 | −13,368.21 | −5593.1 | 1044.96 | 17,263.35 |
| 1990–2000 |  | −15,462.75 | 2330.69 | 121,009.96 | 598 | 1964.66 | −110,353.25 |
| 2000–2010 | Variation (hm²) | −36,534.09 | −30,532.99 | 112,951.36 | −11,555.10 | 58,847.52 | −93,150.09 |
| 2010–2020 |  | −47,527.00 | 30,084.81 | −125,382.01 | 8803.82 | 94,475.19 | 39,329.73 |
| 2000–2020 |  | −84,061.09 | −448.19 | −12,430.65 | −2751.29 | 153,322.72 | −53,820.36 |

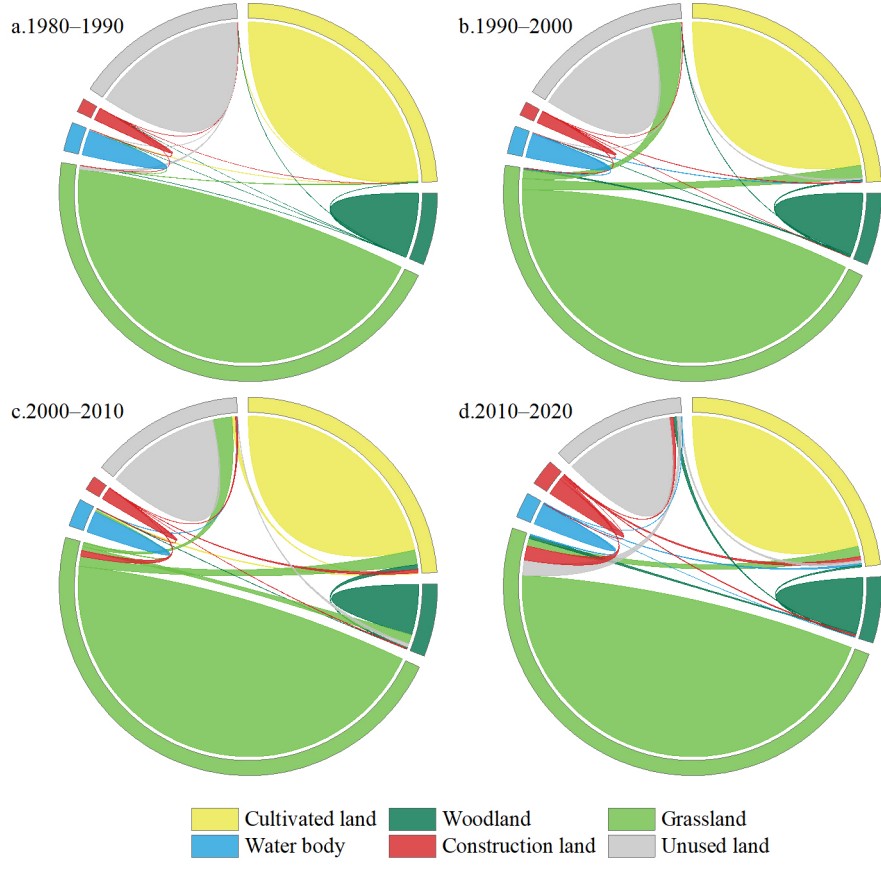

**Figure 3. (a–d)**: Chord diagram of land-use transfer in the SEZ from 1980 to 2020. (The chord diagram is utilized to depict the correlation between multiple land-use types. The line segment that connects two points on a circle is referred to as a chord. Each chord represents the transformation between two land-use types, and the thickness of the chord represents the size of the transferred area.)

## 4.2. Ecological Resilience Spatial and Temporal Patterns

This study calculated the ecosystem resilience index in the SEZ from 1980 to 2020, using grid units, and divided the resilience index into three grades: low (0.00–0.23), medium (0.23–0.35), and high (0.35–0.80) by using the natural breaks (Jenks) method, to analyze the SEZ ecosystem resilience's spatial and temporal changes from 1980 to 2020 (Figure 4).

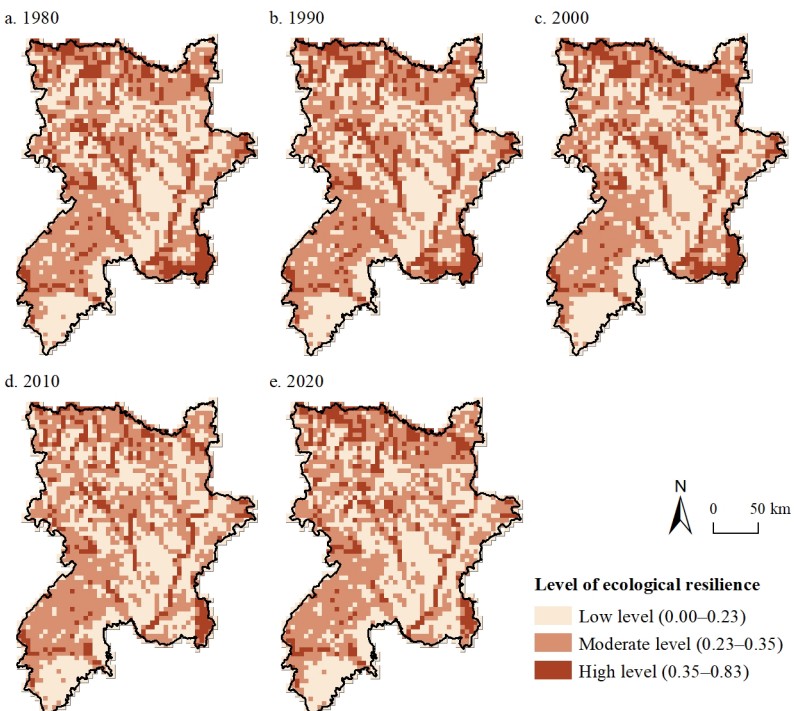

**Figure 4.** (**a–e**): Spatial and temporal changes in the ecological resilience level in the SEZ from 1980 to 2020.

Between 1980 and 2000, there was only a slight decrease in ecological resilience, with the average decreasing from 0.264 to 0.263. The spatial distribution did not change much during this period. However, between 2000 and 2010, the mean value of ecological resilience decreased significantly to 0.256. Specifically, the woodland area in the southern part of Xing County, Shanxi Province, diminished dramatically, leading to a significant decrease in ecological resilience. This is possibly because the region neglected the protection of ecological land while constructing production spaces and mineral resource transportation roads, resulting in a significant shrinkage in ecological land area. From 2010 to 2020, regional ecological restoration projects led to a significant increase in vegetation coverage, and the average ecological resilience value was 0.260 in 2020. The ecological recovery of the Yellow River Basin in the northern part of the SEZ was evident. However, a significant portion of land in the central towns of Shenmu City and Yuyang District in Shaanxi Province has been converted into production and habitable land, resulting in a decrease in ecological land area and an increase in landscape fragmentation. This has led to the formation of a low-value agglomeration area of ecological resilience that spreads outward from the central urban area.

## 4.3. Spatial Relationship between Ecological Resilience and Energy Development

In this study, a kernel density analysis of energy development enterprises in the SEZ was conducted using ArcGIS 10.3 software. In addition, univariate and bivariate local spatial autocorrelation analyses were performed on the regional ecological resilience level and the energy development enterprise kernel density for 2020 using Geoda 1.16 software (Figure 5).

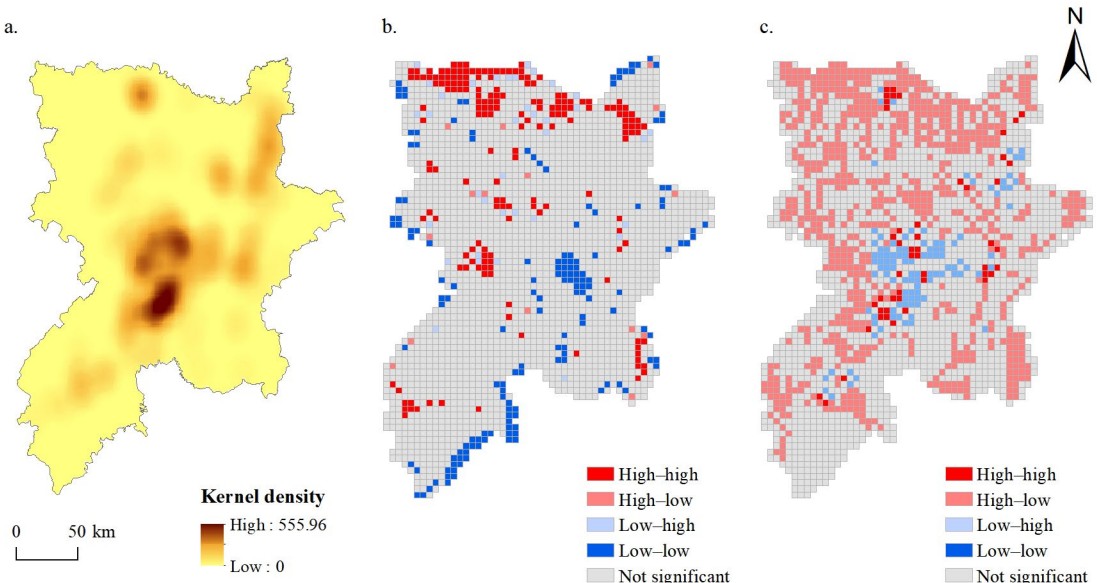

**Figure 5.** Spatial relationship between ecological resilience and energy development in the SEZ in 2020. ((**a**) Kernel density map of energy development enterprises; (**b**) local spatial autocorrelation map of ecological resilience level; (**c**) local spatial autocorrelation map of ecological resilience level—energy development enterprises).

Energy development enterprises were primarily concentrated in the central part of Shenmu City, forming a high-value circle of high kernel density levels spreading outward from the central urban area rich in energy and minerals (Figure 5a). Shenmu City is China's largest coal-producing county (city), with abundant resources and superior mining conditions; thus, the advantages of mineral resources have attracted a huge energy development industry chain and promoted the agglomeration of energy and mineral sewage enterprises.

The local spatial autocorrelation result of the ecological resilience level (Figure 5b) indicates that the areas with high ecological resilience values tended to cluster along the Yellow River in the northern part of the SEZ, illustrating that effective ecological management has significantly enhanced the ecological resilience of the Yellow River Basin in regards to sand and water problems. The bivariate spatial autocorrelation result of the analysis of the ecological resilience level and the kernel density of energy development enterprises (Figure 5c) reveals that energy development enterprises were concentrated in the central part of the SEZ, and there was a spatial clustering distribution of low resilience and high kernel density of energy development enterprises. The clustering of energy development enterprises is likely to have resulted in excessive discharge of wastewater and waste residues generated during production activity, contaminating the ecological environment. Furthermore, the concentration of enterprises has also led to the expansion of construction land, encroaching on ecological land, and decreasing the ecological resilience level in the surrounding areas. Concurrently, regions with a high intensity of human activity and a high risk of pollution where energy development enterprises were clustered were prone to ecological deterioration and a decrease in ecological resilience level, making them focal points that need to be regulated and controlled.

### 4.4. Multi-Scenario Simulation

#### 4.4.1. Land Use under Different Scenarios

Because the SEZ is extensive, the land-use distribution map of the entire area cannot effectively illustrate the differences in land use in various scenarios. Therefore, selected areas within the three localities, along the Yellow River in the northern part of the SEZ, the downtown area of Shenmu City, and the woodlands of Xingxian County, are highlighted to provide detailed insights (Figure 6).

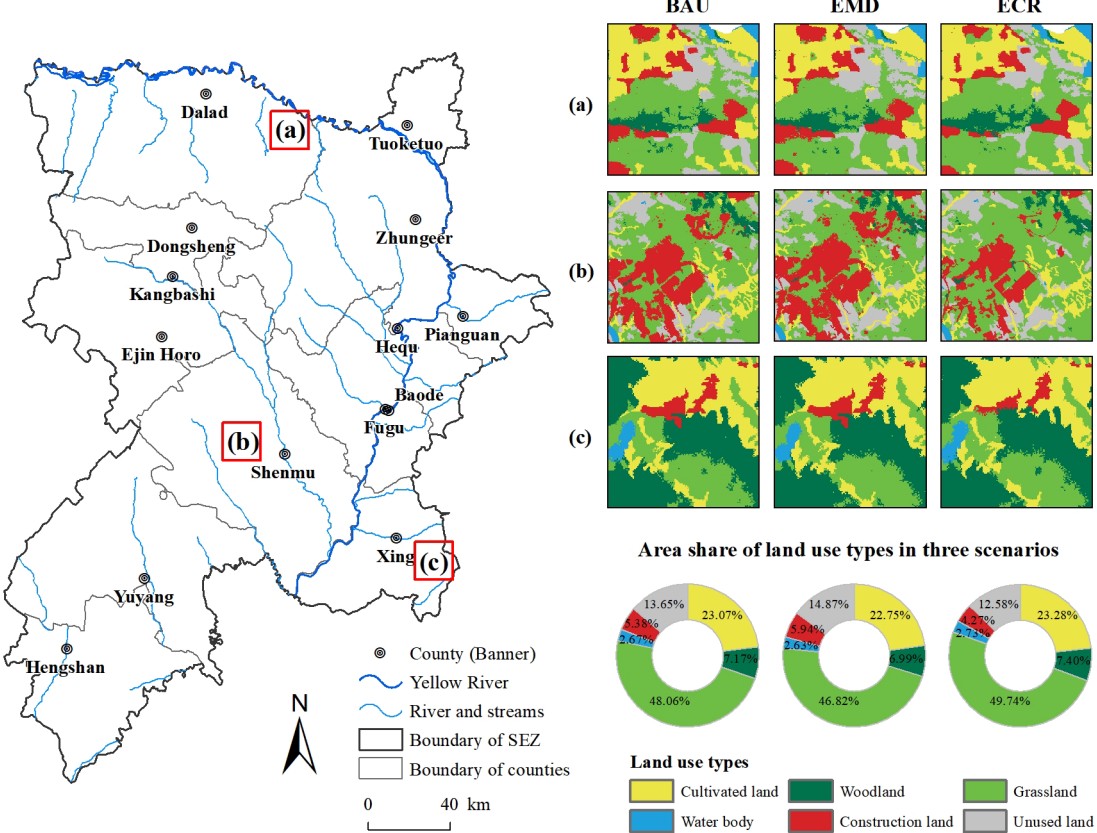

**Figure 6.** Prediction results of land use in the SEZ in 2030 under three scenarios.

In general (Table 4), under the BAU scenario, cultivated land area in the SEZ will decrease by 2.9% in 2030, and grassland area will decrease by 3.8% compared with those areas in 2020. The woodland and water body areas will increase by 8.6% and 6.1%, respectively. This indicates that under the BAU scenario, the SEZ will continue its basic land-use change trend from 2010 to 2020.

**Table 4.** Areas of land use in 2020 and various scenarios in 2030.

| Scenario | Index | Cultivated Land | Woodland | Grassland | Water Body | Construction Land | Unused Land |
|---|---|---|---|---|---|---|---|
| 2020 | | 1,285,290.65 | 357,009.95 | 2,702,636.34 | 136,160.51 | 224,722.55 | 703,839.01 |
| BAU | Area (hm²) | 1,248,048.81 | 387,605.88 | 2,599,639.65 | 144,434.97 | 291,278.43 | 738,682.29 |
| EMD | | 1,230,900.15 | 377,966.50 | 2,533,097.43 | 142,015.19 | 321,362.89 | 804,347.85 |
| ECR | | 1,259,151.26 | 400,574.91 | 2,690,466.04 | 147,884.74 | 230,908.65 | 680,704.40 |

Under the EMD scenario, cultivated land and grassland area will decrease, and construction land area in the SEZ in 2030 will rise significantly by 43% compared to the actual land-use area in 2020, indicating that under this scenario, energy exploitation and related construction activity will occupy the ecological space, exacerbating soil erosion and land desertification problems.

Under the ECR scenario, the woodland and water body areas in the SEZ in 2030 will rise significantly by 12.2% and 8.6%, respectively, compared with those of 2020. The increase in construction land area will be limited to 2.8% compared to 2020. This indicates that under this scenario, ecological land will be protected, construction land expansion will be suppressed, and ecological functions such as regional ecological protection and water conservation will be restored.

### 4.4.2. Ecological Resilience under Different Scenarios

The spatial distribution of the SEZ ecological resilience, resistance, adaptability, and recovery in 2030 under the BAU, EMD, and ECR scenarios is shown in Figure 7.

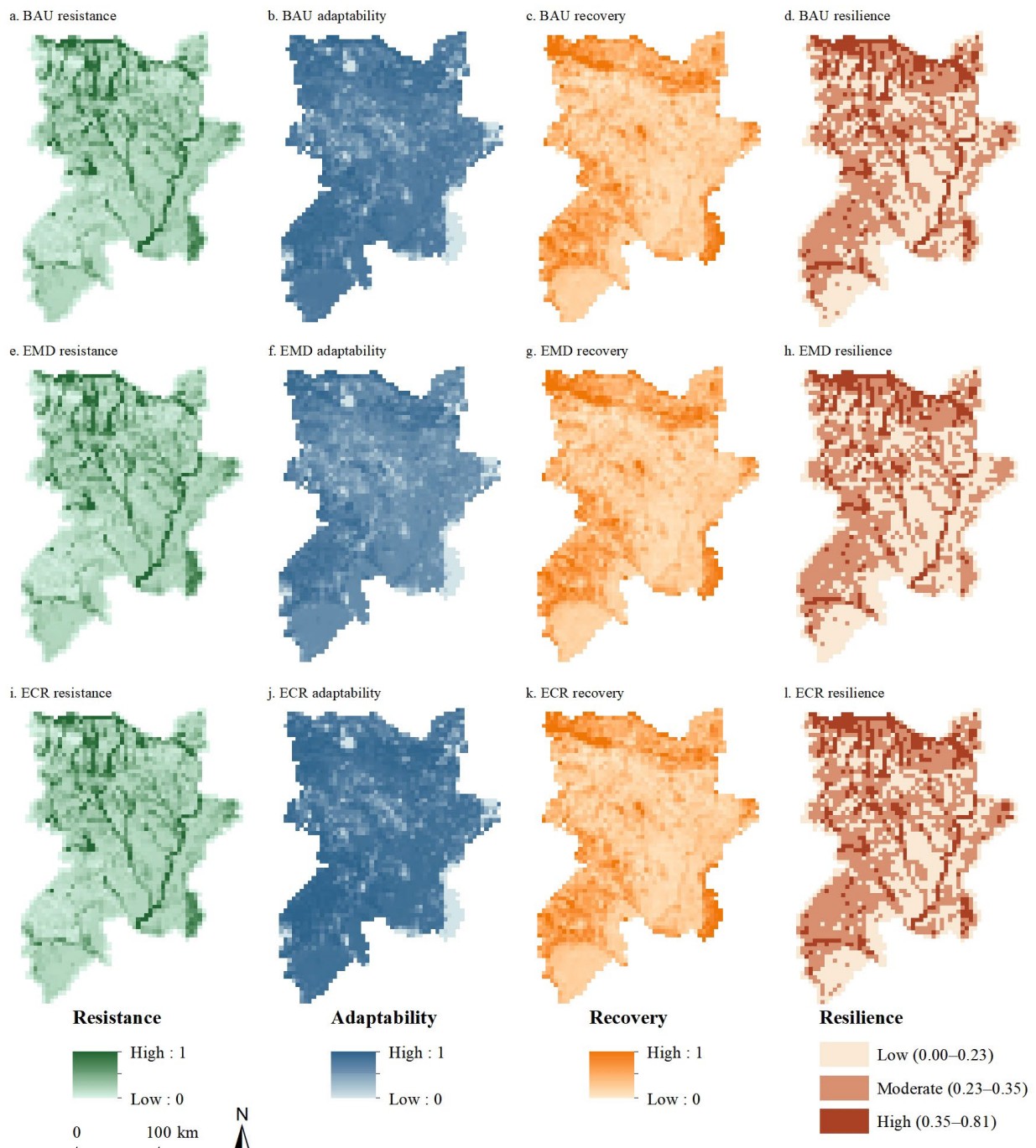

**Figure 7.** Spatial distribution of ecological resistance, adaptability, recovery, and resilience in the SEZ in 2030 under three scenarios. ((**a–d**) The BAU scenario; (**e–h**) the EMD scenario; (**i–l**) the ECR scenario).

1.  BAU scenario. The proportions of low, medium, and high ecological resilience under the BAU scenario are 40.2%, 44.7%, and 15.1%, respectively. Among them, the proportion of low-level resilience increases by 1.8% compared to that of the actual situation in 2020, indicating that if not restricted, the SEZ's ecological resilience will decrease

further. The areas with high ecological resilience and resistance are located in the Yellow River Basin in the northern part of the SEZ, and the low-value adaptability area is located in the woodlands of the eastern part of Xing County in Shanxi Province. The high-value recovery areas are located in the western part of Yuyang District, Shaanxi Province, where unused land predominates, with less anthropogenic-dominated construction land, and a better recovery capacity;

2.  EMD scenario. The proportions of low, medium, and high ecological resilience are 44.3%, 42.6%, and 13.1%, respectively. Ecological resilience is low in Shenmu City, located in the central part of the SEZ, and Hengshan County, in the southern part of the SEZ. The overall low level of ecological adaptability in the SEZ may be explained by the significant growth of construction land area under this scenario, resulting in landscape fragmentation and reduced landscape connectivity;

3.  ECR scenario. The proportions of low, medium, and high ecological resilience are 37.5%, 46.4%, and 16.1%, respectively. Areas with high ecological resilience and resistance are mostly concentrated in the Yellow River Basin and the eastern part of Xing County, Shanxi Province. In addition, the overall ecological adaptability is high, indicating that under the ECR scenario, the SEZ emphasizes ecological land protection, reduces construction land encroachment on ecological land, and reduces landscape fragmentation. This will increase landscape heterogeneity and connectivity, and there will be a high level of ecological resilience in the future.

## 5. Discussion

### 5.1. Expansion Trend of Construction Land in the SEZ

To analyze the expansion trend of construction land in the SEZ more specifically, this paper divided the construction land into three types: urban land, rural land, and mining and transportation land, based on the classification rules of the land-use data used. Furthermore, the paper simulated the land use in 2030 under the BAU scenario. The expansion trends for the three land-use types are presented in Figure 8 from 2000 to 2030, as the changes from 1980 to 2000 were not apparent.

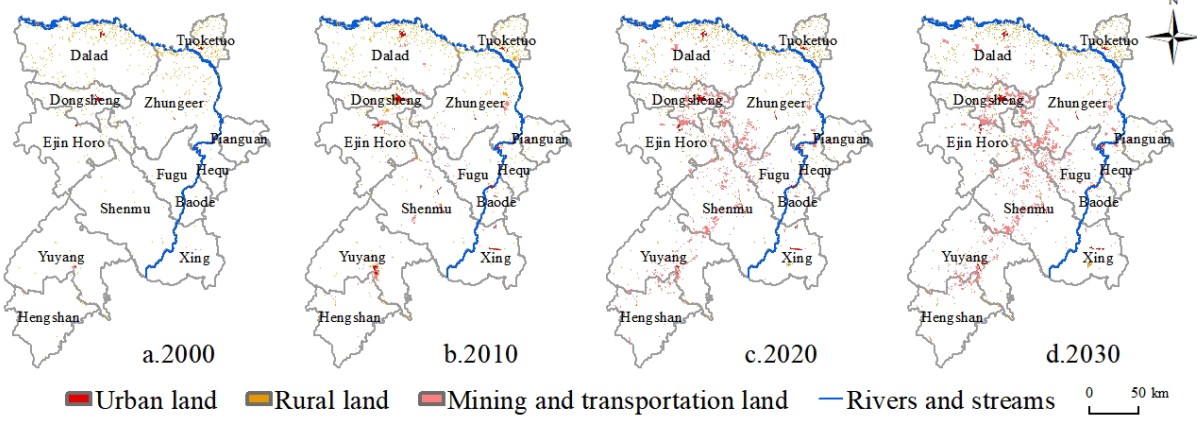

**Figure 8.** (**a**–**d**): Evolution of the three types of construction land between 2000 and 2030.

Since the 21st century, China's diversified development has brought about dramatic shifts, including changes in the spatial patterns of construction land expansion [53]. As is shown in Figure 8, from 2000 to 2010, the construction land in the SEZ was characterized by a dominance of rural land, primarily concentrated along the Yellow River in the northern areas. Meanwhile, during this period, there was a significant increase in urban land, amounting to 10,476.21 hm², which was clustered in the urban centers of Dalad Banner, Dongsheng District, and Yuyang District. This shift indicates that the acceleration of urbanization in the SEZ has led to the expansion of urban land. In 2020, the most notable feature was the rapid growth of mining and transportation land use, which became the dominant

construction land in the region. Mining and transportation land was mainly concentrated in the central part of the SEZ, as well as in Dongsheng District and Shenmu City. Both of these areas are important coal resource bases in China, and energy development activities drive the rapid expansion of mining and transportation land use. According to the simulated 2030 BAU scenario, the expansion of mining and transportation land will continue and dominate the regional construction land in the Coal Resource Zone. In addition, as can be seen in Figure 9, the overall area of construction land in the SEZ grows rapidly between 2000 and 2030; however, this growth is accompanied by a rapid decline in cultivated land and ecological land (woodland, grassland, water body, and unused land), which will pose a significant threat to the ecological resilience of the region.

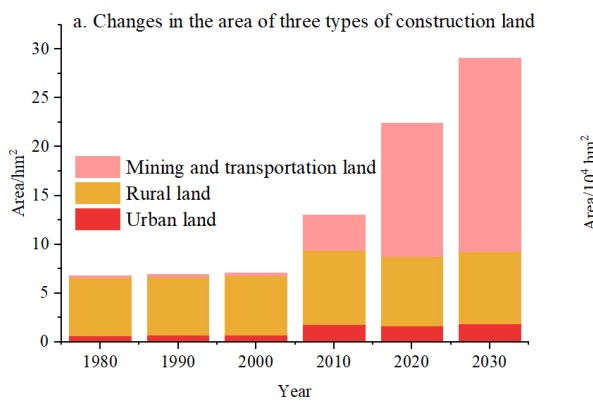 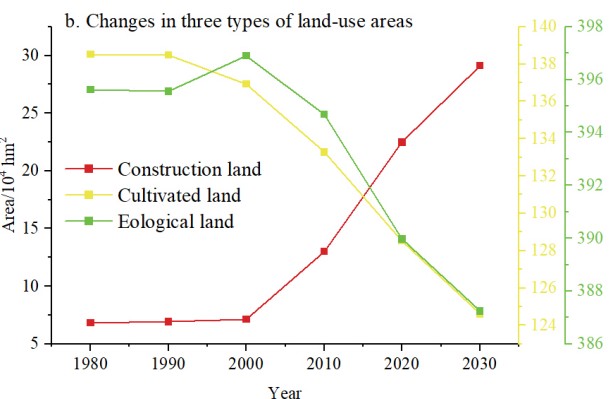

**Figure 9.** Evolution of different types of land use between 1980 and 2030. ((**a**) Changes in the area of mining and transportation land, rural land, and urban land; (**b**) Trends in the area of construction land, cultivated land, and ecological land).

Energy development projects and construction activities within energy development zones play a crucial role in driving regional economic growth. However, as the analysis presented above shows, it is evident that such resource development activities have adversely impacted the regional ecosystem. The expansion of construction land, mainly for mining and transportation, has caused a decline in the region's ecological resilience. China's economic progress has been challenged by the need for ecological conservation, especially in energy development zones. Balancing energy extraction, economic growth, and ecological protection has led to a focus on the gradual allocation of construction land as a potential solution to this dilemma. Based on the simulated results for 2030, it is expected that the main land-use change within the SEZ will be the continuous growth of construction land, driven by energy development. In light of this, the SEZ may need to prioritize the effective use of construction land, rather than solely focusing on controlling its growth. For instance, implementing comprehensive management strategies for coal mining subsidence areas and strengthening land reclamation efforts can help to effectively restore the use of construction land resources.

### 5.2. Comparison of Multi-Scenario Ecological Resilience Simulation Results

Compared to the BAU scenario, the ecological resilience in the SEZ in 2030 under the EMD scenario declines significantly, with a 4.1% increase in the low-level proportion and a 2% decrease in the high-level proportion. This suggests that energy extraction must be accompanied by a focus on protecting and restoring ecosystems. In the eastern part of the SEZ, the ecological resilience of Pianguan, Baode, and Xing counties in Shanxi Province declines with the shrinkage of woodland and grassland areas, illustrating that the construction land area encroaches upon ecological land during production and construction activity. Furthermore, the ecological adaptability in the EMD scenario is significantly lower. This could be attributed to the substantial increase in construction land area under the EMD scenario, leading to sporadic accumulation and infiltration into other land-use types. This,

in turn, causes landscape fragmentation and reduces landscape connectivity. Therefore, the energy zones must take decisive measures to prohibit private mining of iron ore resources and other activities that have a detrimental impact on the environment. It is imperative to adopt and implement advanced technology to modernize traditional mining practices, promote sustainable mining, and enhance mineral resource recovery rates and comprehensive utilization rates. Furthermore, it is necessary to reduce the dependence on highly polluting resources and ensure a sustainable supply of energy and raw materials for various industries.

Under the ECR scenario, the ecological resilience in 2030 is higher than that in the BAU and EMD scenarios, with a high level of ecological resilience accounting for 16.1% of the area. This shows that the woodland and water bodies in the SEZ are effectively protected under the ECR scenario, resulting in an increase in ecological resilience in woodlands and along water bodies. Ecological conservation projects are essential for protecting regional ecology, improving ecological quality, enhancing ecological resilience, and promoting sustainable regional development. The SEZ is a key area for the ecological protection and restoration of the Yellow River Basin. In March 2000, the State Council implemented a policy of returning cultivated land to woodland (grassland) to increase vegetation cover and control soil erosion. In this context, the SEZ has successfully implemented ecological projects such as the construction of silt embankments and terraces, the "Three Norths" protective forests, and Yellow River Basin water loss and soil erosion comprehensive management, effectively reducing the degree of erosion of the regional slopes and channels, further promoting regional ecological restoration, and improving the ecological resilience. Additionally, the ecological resistance and adaptability in the ECR scenario are higher, indicating that construction land expansion in the ECR scenario is effectively suppressed and the ecological land is restored, which contributes to the enhancement of ecosystem service functions and a decrease in landscape fragmentation in the SEZ.

The above analyses demonstrate that in energy zones like the SEZ, ecological conservation is essential for improving regional ecological resilience. In 2017, the concept of "high-quality development" introduced by the state shifted from traditional rough development to prioritizing green development for the harmonious coexistence of humans and nature. Therefore, it is crucial to prioritize ecological principles and pursue moderate development in actual development efforts. This may involve focusing on the development of ecological industries that align with the primary goal of ecological conservation, particularly in the counties and central towns within the region. Additionally, ongoing efforts to enhance ecological construction should be prioritized to ensure the integrity of the ecosystem is maintained to the fullest extent possible.

*5.3. Research Insights*

Regarding the SEZ, ecological resilience is generally low in the intensive areas of energy development enterprises and their surroundings, indicating that industrial production activities such as energy extraction have seriously affected the surrounding land-use types. Extraction of energy and mineral resources is temporary [54]; therefore, it is necessary to strengthen land restoration in these areas by comprehensively integrating and rehabilitating land that has been damaged by energy and mineral extraction. In 2011, the State Council issued the Regulations on Land Reclamation, which required the remediation of land destroyed by production and construction activity to make it available for use. Under the EMD scenario established in this study, construction and unused land areas increased by 43% and 14.3%, respectively, compared with those of 2020. Thus, ecological restoration of industrial and abandoned energy and mineral development production land is required to reduce landscape fragmentation caused by the occupation of ecological land for production and construction to improve regional ecological resilience. Moreover, governments should set up and implement the "Who Destroys, Who Compensate" concept of ecological compensation in energy-rich regions, to make full compensation for the economic and societal

losses. Moreover, it is essential to enhance the efficiency of productive and habitable land to mitigate the adverse effects of construction land expansion [55,56].

It is a fact that mining activities cause environmental disturbances. Nevertheless, it is necessary to take into account the evolution of the Chinese economy and to observe the rules of economic and social development. Given that coal remains a primary component of China's energy structure, and electricity plays a central role, it is imperative to pursue a balanced approach that integrates protection and development. This entails enhancing the level and efficiency of resource utilization. The results of this paper underscore the importance of ecological conservation in preserving the ecological resilience of energy zones. However, relying solely on technical governance measures may not be sufficient to establish a resilient ecological civilization in these zones. Higher government authorities must implement management measures that guarantee the adoption of ecological conservation and governance techniques. This can be accomplished by incentivizing social funds to invest in ecological conservation within energy zones and promoting ecological restoration in mineral and resource-based areas through preferential policies, including tax incentives and financial subsidies.

### *5.4. Shortcomings and Prospects*

First, in Section 3, this paper chose the simpler method of ESV to measure ecological resistance, which may have some limitations. Due to the complexity of the economic valuation of ecosystem services, the methodology of ESV has been questioned in practical applications [57,58]. Criticisms include concerns regarding the mechanistic nature of value assignment and the unilateral nature of evaluation criteria. Research on the ecological significance of ecosystem services may be neglected due to excessive focus on calculating their economic value. For example, the economic value of unused land is difficult to account for, which may result in a portion of the ecological value of its regulation and support not being captured. In addition, ecosystem change is a dynamic process, and the area-weighted approach to calculating the total value of ecosystem services may cause functional de-differentiation and impede the identification of faster-changing ecosystem services [42], leading to less accurate construction of ecosystem resistance.

Second, in the previous study, to ensure that the results of the simulation are comparable to the actual land use, we classified the land use into six types, which may limit the reliability of specific land-use types. For example, previous research [59] classified open-pit coal mines as a land-use type based on specific research needs, which can improve the analysis accuracy of coal mine land-use types. Although we discussed the refinement and modeling of the construction land, we did not proceed to calculate the ecological resilience of the refined land types due to the indicator calculation method chosen. However, we believe that the different types of construction land have varying levels of resistance, recovery, and adaptability, and this should serve as a basis for future research to investigate the impact on ecological resilience after refining the land-use types. Furthermore, the development of energy zones was not limited to the three scenarios set in this study. In the future, we can consider scenarios that coordinate economic development and ecological protection, seeking a balance between the two, rather than just considering one aspect. Simulation results can better support planning policies when the scenario setting is coupled with multi-objective optimization functions.

### 6. Conclusions

This study analyzes land-use change characteristics of the SEZ from 1980 to 2020, explores the spatial and temporal variation laws of regional ecological resilience, reveals the spatial agglomeration pattern of regional ecological resilience and energy development enterprises and their correlations, and finally simulates the spatial pattern of land use and ecological resilience of the SEZ in 2030. The results show that the land-use type of the SEZ is dominated by cultivated land, grassland, and unused land, and the construction land area increased dramatically in the 21st century. The ecological resilience level in the SEZ

showed a trend of first decreasing and then increasing. From a spatial perspective, the high ecological resilience value area was distributed in a belt shape in the Yellow River Basin, whereas the central urban area of the city spread outward to form a low ecological resilience agglomeration area. The ecosystem resilience level was high in southeastern Xing County, Shanxi Province, whereas the ecological resilience level was low in Fugu County, Shenmu City, and Hengshan County, Shaanxi Province. The expansion of energy-based enterprises and resource-based cities drove construction land expansion, resulting in generally low ecological resilience in these areas.

Although the BAU scenario showed an increasing trend in woodland and water bodies, the decrease in regional ecological resilience was still mainly due to the expansion of the construction land area. The increase in construction land area under the EMD scenario was particularly prominent, leading to a low level of ecological resilience, which does not support sustainable regional development. However, the ECR scenario was more conducive to improving ecological resilience and realizing sustainable development in the SEZ. Therefore, during energy zone development, it is necessary to focus on ecological environmental conservation and sustainable ecosystem management to continuously improve regional ecological resilience.

**Author Contributions:** Conceptualization, X.C.; Formal analysis, X.C.; Funding acquisition, Y.S.; Methodology, X.C.; Supervision, Y.S., D.X., B.M., X.L. and L.Z.; Writing—original draft, X.C.; Writing—review and editing, Y.S. and D.X. All authors have read and agreed to the published version of the manuscript.

**Funding:** This research was funded by the National Natural Science Foundation of China, grant number 42001251; the Fundamental Research Funds for the Central Universities, grant number GK202201008.

**Data Availability Statement:** The data presented in this study are available on request from the corresponding author. The data are not publicly available due to the data is designed to be used in other ongoing research and should be protected before official publication.

**Conflicts of Interest:** The authors declare no conflicts of interest.

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
