# Peer review of "Spatial and Temporal Changes in Ecological Resilience in the Shanxi–Shaanxi–Inner Mongolia Energy Zone with Multi-Scenario Simulation"

_land, doi:10.3390/land13040425_

Round 1
Reviewer 1 Report
Comments and Suggestions for Authors
1) The introduction should provide the state in the literature of the art with respect the topic of the study. How to approach the behavior of the ecosystem should be one part of the introduction because it is directly related to the topic studied. You stated: How can regional ecosystems to be measured? The reader one finished with your introduction will not able to responde at the question you asked. To cover your question, you should provide the different approachs available for such matter, are differences among these approaches, what approach did opt for and why.
2) In the section method, you provided a certain amount of litterature for the methods you used (resistance, adaptability and recovery). I think most of that litterature should move to the Introduction to reinforce the question you asked about the measurements of the regional resilience. The section method should be restricted to just providing and describing the methods.
3) For assessing changes in systems with time, 20 years is kind of insufficiant. It would be better to increase it. For instance, How was the area before 2000 and how the area of the Socio Ecological System studied evolved with time during the last five or sixe decades.
4) For the discussion of the senarios, you should take into consideration the policy implemented in the past and the one of the futur.
Author Response
Dear Editor(s) and Reviews:
We would like to thank you very much for your valuable comments and good suggestions that greatly helped to improve our manuscript titled “Spatial and temporal changes of ecological resilience in the Shanxi–Shaanxi–Inner Mongolia energy zone with multi-scenario simulation”. We have carefully considered your valuable comments and good suggestions. We used the revision mode of the Word document to make changes to the article. In addition, we explained how your comments have been taken into full account in the revision notes, and the comments are responded point by point below.

Reviewer 2 Report
Comments and Suggestions for Authors
Manuscript presents a very interesting and actual topic. It is quite innovative in an attempt to address the issue of ecological resilience based on resistance, adaptation and recovery indicators. The paper is well balanced and comprehensive. Yet, there are still few issues that in my opinion deserve clarification or deeper justification.
Energy development zones – this expression needs a deeper explanation or description for an international reader to better understand all its nuances. In case, it is a legal term in China, please provide its definition or at least its description.
Another issue is the land use category “construction land” that as I understood comprises both built-up areas (residential and industrial) and mines/quarries/oil extraction sites. This is a principal problem when assessing ecological resilience since all these sub-categories have different capacity for resistance, adaptation or recovery…as the result of their ecological structure (mixture of natural/artificial ecosystems in urban residential area vs. sparse or none ecosystems in open mines, quarries or dump sites).
Please, explain in detail how ESV is related to ecological resistance and justify why EVS may be used for resistance index. For example, usually unused land has a high ecological value and provide many regulatory and supporting (for biodiversity and nature conservation) services while having low economic value. This is somewhat not reflected in the ESV values used in your manuscript (table 2).
In this respect, I suggest to change colours in Figure 6 (use standardised colour schemes for land use, use colour for cultivated land and split settlements from industrial/mineral extraction sites).
I also suggest to highlight the main differences in the spatial distribution in resilience in Figure 7 in order to show how and where those 3 scenarios affect the indices most.
Author Response

(The authors gave the same response as above.)

Reviewer 3 Report
Comments and Suggestions for Authors
I have carefully reviewed the manuscript entitled "Spatial and temporal changes of ecological resilience in the Shanxi-Shaanxi-Inner Mongolia energy zone with multi-scenario simulation" and would like to provide my assessment. Below you will find my assessment of the strengths and weaknesses of the manuscript:
Commendable points:
Timeliness of the topic: the topic of the manuscript is highly relevant as it addresses spatial and temporal changes in ecological resilience in a major energy zone. Given the increasing importance of environmental aspects in the energy sector, this topic is highly relevant for the scientific community.
Clarity of the methodological approach: The authors have presented the methodological approach clearly and comprehensibly. By using multi-scenario simulations, they provide an insight into the potential impacts of different developments on ecological resilience in the region under investigation.
Criticisms:
Relevance for an international audience: There is a lack of clear presentation of the relevance of the results for an international audience. Although the topic itself could be of global relevance, it is not made clear why the findings from this specific region could also be of interest to other regions or to global research.
Critically questioning the concept of ecosystem services: The concept of ecosystem services needs to be critically scrutinized in terms of its appropriateness, especially in light of the functional de-differentiation it entails. The authors should deepen their argumentation in this area and discuss possible limitations or problems in the application of this concept.
Clarity of figure and form of presentation: Figure 3 in the manuscript can be described as aesthetically pleasing, but it is not really clear what information it is intended to convey. It is recommended to choose an unambiguous form of presentation that presents the results of the study clearly and comprehensibly.
Overall, the manuscript makes an important contribution to research into ecological resilience in a specific energy zone. However, the authors should consider the points of criticism mentioned and expand and clarify their argumentation accordingly.
Author Response

(The authors gave the same response as above.)

Round 2
Reviewer 3 Report
Comments and Suggestions for Authors
Thank you, I agree with your revision.